# Ontological Analysis of Coronavirus Associated Human Genes at the COVID-19 Disease Portal

**DOI:** 10.3390/genes13122304

**Published:** 2022-12-07

**Authors:** Shur-Jen Wang, Kent C. Brodie, Jeffrey L. De Pons, Wendy M. Demos, Adam C. Gibson, G. Thomas Hayman, Morgan L. Hill, Mary L. Kaldunski, Logan Lamers, Stanley J. F. Laulederkind, Harika S. Nalabolu, Jyothi Thota, Ketaki Thorat, Marek A. Tutaj, Monika Tutaj, Mahima Vedi, Stacy Zacher, Jennifer R. Smith, Melinda R. Dwinell, Anne E. Kwitek

**Affiliations:** 1The Rat Genome Database, Department of Biomedical Engineering, Medical College of Wisconsin, Milwaukee, WI 53226, USA; 2Clinical and Translational Science Institute, Medical College of Wisconsin, Milwaukee, WI 53226, USA; 3Finance and Administration, Medical College of Wisconsin, Milwaukee, WI 53226, USA; 4The Rat Genome Database, Department of Physiology, Medical College of Wisconsin, Milwaukee, WI 53226, USA

**Keywords:** COVID-19, database, disease enrichment, liver disease, biological knowledgebase, gene set enrichment

## Abstract

The COVID-19 pandemic stemmed a parallel upsurge in the scientific literature about SARS-CoV-2 infection and its health burden. The Rat Genome Database (RGD) created a COVID-19 Disease Portal to leverage information from the scientific literature. In the COVID-19 Portal, gene-disease associations are established by manual curation of PubMed literature. The portal contains data for nine ontologies related to COVID-19, an embedded enrichment analysis tool, as well as links to a toolkit. Using these information and tools, we performed analyses on the curated COVID-19 disease genes. As expected, Disease Ontology enrichment analysis showed that the COVID-19 gene set is highly enriched with coronavirus infectious disease and related diseases. However, other less related diseases were also highly enriched, such as liver and rheumatic diseases. Using the comparison heatmap tool, we found nearly 60 percent of the COVID-19 genes were associated with nervous system disease and 40 percent were associated with gastrointestinal disease. Our analysis confirms the role of the immune system in COVID-19 pathogenesis as shown by substantial enrichment of immune system related Gene Ontology terms. The information in RGD’s COVID-19 disease portal can generate new hypotheses to potentiate novel therapies and prevention of acute and long-term complications of COVID-19.

## 1. Introduction

Coronavirus Disease 2019 (COVID-19) was declared a global pandemic in March 2020 and now, two years later, the disease still plagues the globe due to rapid mutations within the causal virus, SARS-CoV-2, and inadequate preventive measures. Individuals infected with the virus exhibit a wide spectrum of symptoms, from asymptomatic infection to acute respiratory distress requiring hospitalization [1]. The major manifestation of COVID-19 is in the respiratory system, where the virus infects nasal mucosa and spreads into the host body [2,3]; however, non-respiratory systems such as liver, heart, kidney and brain, are also involved in certain patients and, in severe cases, result in multiple organ failure and death [4]. ACE2, the receptor for SARS-CoV-2, has a broad distribution in tissues such as blood vessels [5], small intestine, heart, kidney, thyroid, adipose and testis [6]. These ACE 2 expressing organs become targets of SARS-CoV-2 infection. Once entering the cell, the binding of viral spike protein to ACE2 initiates signaling pathways promoting inflammatory mediator production [7]. Other non-ACE2 mediated infection might involve viral entry through nerve endings like olfactory nerves to the central nervous system causing neurological symptoms [8]. In addition to direct viral damage to the target organs, dysregulation of cytokine release, referred as cytokine storm, plays a role in the severity of the disease. For example, the pre-existing cytokine dysregulation in diabetic patients amplified the cardiovascular damage resulting from COVID-19 [9] and poor disease prognosis [10]. Dysregulation of cytokine secretion causes a severe inflammatory response in organs leading to multiple organ failure [4] or contributes to long COVID even after patients recovered from COVID-19 [11,12].

The vaccine development for COVID-19 has been a successful example of moving from knowledge to saving lives. So far, 11 vaccines have been approved by the World Health Organization (WHO) that are able to protect people from infection or severe disease outcome (https://www.who.int/emergencies/diseases/novel-coronavirus-2019/covid-19-vaccines (accessed on 1 June 2022)) [13]. However, the frequently changing viral genome demands development of therapeutics as another solution to control the pandemic. Remdesivir, approved for treatment of hepatis C originally, was the first antiviral treatment of COVID-19 approved 10 months after identification of SARS-Co2 virus [14]. To meet the urgent need, several repurposed drugs were on clinical trials and recommended for COVID-19 under specified conditions [15].

Both basic and clinical research is progressing at an unprecedented rate to tackle the COVID-19 pandemic and the long-term health effects of SARS-CoV-2 infection, resulting in 265,064 COVID-19-related publications in PubMed and 23,621 unreviewed COVID-19 preprints (June 2022). Since COVID-19 is an emerging disease, the surge in original publications is complemented by numerous reviews and large-scale meta-analyses to present organized knowledge of COVID-19 to the public. In just over two and a half years, there are more than 3,000 review articles focusing on COVID-19 in PubMed. Leveraging this vast knowledge and applying adequate standards to integrate COVID-19 data from different fields is challenging yet at the same time critical to accelerate the ongoing development of effective preventive measurements and new therapies.

The Rat Genome Database (RGD; rgd.mcw.edu is a cross-species database targeting specific disease areas for its ongoing manual disease curation [16]. During early 2020, the RGD curation team launched an infectious disease curation project with an initial focus on the coronavirus infectious diseases. The COVID-19 Portal and the Infectious Disease Portal were released sequentially to present integrated data resources to the research community. Here, we introduce the COVID-19 Disease Portal, that was released within six months of the pandemic, and present results analysis on COVID-19 associated genes using a series of RGD resources and tools [17] in order to visualize organ distribution of COVID-19 genes and their relationship with other diseases. The Gene Annotator [18] tool, which provides counts of query genes that are associated with other diseases across organ systems, based on their annotations, was used to serve this purpose. Disease ontology enrichment analyses examined which disease terms are over-represented in the annotations. Two enrichment tools: Multi Ontology Enrichment Tool, MOET [19] from RGD, and Set Analyzer [20] from CTD (Comparative Toxicogenomics Database) were used to find enriched disease terms and to compare the results between the tools.

We first examined the disease gene distribution among organ system diseases using the Gene Annotator tool and found that over sixty percent of COVID-19 genes were also associated with “nervous system disease.” Both MOET and Set Analyzer identified liver diseases as the significantly enriched disease terms on the COVID-19 associated gene list. To further understand the molecular mechanism involved in COVID-19 pathogenesis, we examined Gene Ontology (GO) annotations enrichment of these genes. Both biological process and molecular function annotations are highly enriched in the immune-associated branches which confirm the involvement of immune dysregulation among COVID-19 patients.

## 2. Materials and Methods

Targeted curation of literature related to coronavirus infection was performed at RGD using the in-house curation tool [18] integrated with the OntoMate [21] literature searching tool. A standalone OntoMate tool is also accessible at https://rgd.mcw.edu/QueryBuilder/. The prioritized disease gene list was constructed as previously described [22] with added COVID-19-related genes from other sources such as the Gene Ontology Consortium (GOC) (http://geneontology.org/covid-19.html (accessed on 1 June 2020)) and LitCovid (https://www.ncbi.nlm.nih.gov/research/coronavirus/ (accessed on 1 May 2020 )). In brief, three data bases, MalaCards (https://www.malacards.org/ (accessed on 1 May 2020)), DisGeNET (https://www.disgenet.org/ (accessed on 1 May 2020)) and PhenoPedia (https://phgkb.cdc.gov/PHGKB/startPagePhenoPedia.action (accessed on 1 May 2020)) were queried for coronavirus disease, related viral diseases, and other infectious diseases. The purpose of the queries was to find human genes associated with those diseases in the biomedical literature. A prioritized list of genes was made based on appearance of the genes in multiple databases and the combined number of publications connected to each gene-disease association across those databases. A small subset of unique coronavirus disease-related genes not found in multiple databases was curated first, before curation of the main infectious gene list began. The gene symbol and the disease term “coronavirus infectious disease” was used to find publications associated with coronavirus disease. Using ontological approaches, OntoMate retrieves publications tagged with the coronavirus infectious disease, and any of its child terms such as COVID-19, Middle East respiratory syndrome, severe acute respiratory syndrome…, etc. The resulting publication list was ranked by relevance or sorted by publication dates. In the curation process the relationship between a gene and a disease is indicated by evidence codes [23]. The evidence code IDA (inferred from direct assay) is used to indicate direct involvement of a gene product in causing or treating a disease. IMP (inferred from phenotype manipulation) is used in cases where gene expression/function is artificially altered and a genetic or mechanistic connection between a disease is implied. IAGP (Inferred by Association of Genotype from Phenotype) is used in an association of a disease with genetic mutations or polymorphisms of a gene. IEP (Inferred from Expression Pattern), or HEP (expression changes measured by high throughput assays) is used when a gene changes its expression pattern during the disease course. In addition to in-house manual annotations, RGD regularly imports annotations from other data resources, including the Gene Ontology Consortium, Clinvar, Mouse Genome Informatics, Online Mendelian Inheritance in Man, Online Mendelian Inheritance in Animals and the Comparative Toxicogenomic Database (CTD) [22]. The evidence codes of imported annotations are assigned with the same criteria except for EXP evidence codes used in the annotations imported from CTD. EXP indicates a gene may be a biomarker of a disease or play a role in the etiology of a disease. RGD also propagate annotations from other organisms to human orthologous genes by using the ISO (Inferred from Sequence Orthology) evidence code. These imported annotations are organized into categories such as ‘Disease’, ‘Human Phenotype’, ‘Mammalian Phenotypes’ for non-human organisms, and others as shown in the COVID-19 Disease Portal (Figure 1A).

The genes derived from the COVID-19 Disease Portal were further evaluated using tools available at RGD. The Gene Annotator tool retrieves all functional annotations for a gene list or a chromosomal region and visualizes the gene count distribution across disease terms in a Comparison Heat Map (https://rgd.mcw.edu/rgdweb/ga/start.jsp (accessed on 1 June 2022)). Two publicly available enrichment tools, MOET (the Multi Ontology Enrichment Tool (https://rgd.mcw.edu/rgdweb/enrichment/start.html (accessed on 1 June 2022)) at RGD, and Set Analyzer (http://ctdbase.org/tools/analyzer.go (accessed on 1 June 2022)) at the CTD were used to perform enrichment analysis of COVID-19 disease genes. Both tools are web-based analysis tools that generate a list of ontology terms statistically over-represented with the input gene symbol list. The Set Analyzer finds enriched disease, Gene Ontology, pathways, and gene-gene interaction terms for human genes while MOET is capable of performing enrichment analyses in multiple species (including rat, mouse, human, bonobo, squirrel, dog, pig, chinchilla, naked mole-rat and vervet) and multiple ontologies (including Disease, GO, Pathway, Phenotype, and Chemical entities (ChEBI)). The Ancestor chart from QuickGO (https://www.ebi.ac.uk/QuickGO/ (accessed on 1 June 2022)) [24] was used to visualize the relationship among enriched GO terms.

## 3. Results

### 3.1. The COVID-19 Disease Portal

The landing page of the COVID-19 Disease Portal (https://rgd.mcw.edu/rgdweb/portal/home.jsp?p=14 (accessed on 1 June 2022)) (Figure 1) provides accesses to all the integrated COVID-19 data. The disease browser (Figure 1B) links to the Annotations page where annotations can be downloaded for analysis. The annotations associated with COVID-19 and its child terms were downloaded from the Annotations page and the associated genes were sent to the Gene Annotators and MOET tool [19] for further analysis. The disease gene list can also be obtained from the OLGA tool (https://rgd.mcw.edu/rgdweb/generator/list.html) using the target disease term as a key word. The “Gene Set Enrichment” section (Figure 1C) at the bottom of the page sends genes curated with the highlighted ontology term (COVID-19) to the enrichment tool MOET for analysis. Seven ontologies are available in MOET for enrichment analysis.

### 3.2. Human COVID-19-Associated Gene Analysis

COVID-19 is a member of the coronavirus infectious disease family. This family includes ‘Middle East respiratory syndrome’ (DOID:0080642) (MERS) and ‘severe acute respiratory syndrome’ (DOID:2945) (SARS) (Figure 1B). There are 1257 human genes associated with COVID-19, 19 genes associated with MERS and 90 genes associated with SARS, totaling 1338 coronavirus infectious disease genes at RGD (accessed on 1 June 2022). Their overlapping coverage is visualized in the Venn diagram in Figure 2. The intensity of COVID-19 disease research is reflected by more than one thousand disease related genes identified in just over two years of the pandemic. They comprise more than 90% of the disease genes associated with coronaviruses.

### 3.3. Gene Disease Association

Most of the COVID-19 annotations were identified by IEP and HEP evidence codes. They account for more than 95% of the total 1407 COVID-19 annotations. The COVID-19 annotations are associated with 1257 unique genes (Table 1A). Among these genes, only ACE2 is associated with four types of evidence codes: IAGP, IDA, IMP and EXP, most of them (1221 genes) are with one type of evidence code (Table 1B).

These COVID-19 associated genes are also involved in other diseases as viewed from the annotation distribution heatmap in the Gene Annotator tool (Figure 3A). More than half (703) of the COVID genes are associated with ‘developmental disease.’ In the ‘developmental disease’ branch, 650 genes are associated with ‘congenital, hereditary and neonatal disease’ and 334 with ‘neurodevelopmental disorders’ (Figure 3B). There are 911 COVID genes associated with ‘disease of anatomical entity.’ The breakdowns of the anatomical entities associated COVID-19 disease genes are shown in Figure 3C and discussed in the ‘COVID-19 affected organ systems’ section later.

### 3.4. Disease Term Enrichment Analysis

COVID-19 is a disease with a broad spectrum of symptoms, including some atypical symptoms of respiratory diseases like loss of smell, and neurological symptoms [25]. Use of the existing knowledge of how these COVID-19 genes are involved in other human diseases could shed light on the pathogenesis of COVID-19 and facilitate development of therapeutic strategies. To find these relationships, we next looked at the disease enrichment patterns of the COVID-19 disease genes using MOET [19] developed at RGD. Several high-level disease terms such as ‘coronavirus infectious disease’, ‘RNA virus infection’ and ‘viral infectious disease’ are highly enriched since they are parent terms for COVID-19. The enriched disease table was downloaded from MOET, and the top 40 enriched diseases were selected, from ‘respiratory tract infections’ to ‘lung injury’ and were listed in Table 2A. As expected, the term ‘respiratory tract infections’ was on top of the list. Surprisingly, there were several enriched terms in the liver disease branch, including liver neoplasms, hepatobiliary system cancer and others. Additional enriched terms included rheumatic disease, autoimmune disease of musculoskeletal system, allergic disease, pneumonia, and immune/inflammatory diseases of non-respiratory system diseases. The same COVID-19 disease genes were sent to the Set Analyzer, another enrichment tool at CTD, and the top 40-enriched diseases are listed in Table 2B. Most of the enriched diseases in MOET were also enriched in the Set Analyzer, however, some of the ranking orders were shifted. The ‘respiratory tract infections’ was on top of the MOET list while it was ranked 17th on the list from the Set Analyzer. These differences could be attributed to different ways of data integration and two different disease vocabularies used by RGD [22] and CTD [20]. Overall, liver diseases, immune system diseases, autoimmune diseases and respiratory tract diseases were highly enriched in both tools. The enriched list of the Set Analyzer includes more organ system disease terms while there are more granular terms on the MOET list. Using ‘Nervous System Diseases’ (MESH: D009422) as an example, on the MOET enrichment list, ‘autoimmune disease of central nervous system (DOID:0060004)’ is ranked 38th, however, the high level term of its parents ‘Nervous System Diseases’ (MESH: D009422) is ranked 15th on the enrichment list from Set Analyzer. On the MOET list several granular kidney disease terms such as nephritis, glomerulonephritis, and glomerular diseases are ranked 14th, 17th, and 21st, respectively, while on the Set Analyzer list, ‘Urologic Diseases’ (MESH: D014570) (parent of kidney diseases) is ranked 26th and ‘Nephritis’ (MESH: D009393) 32nd.

### 3.5. COVID-19 Affected Organ Systems

We now look at the target organ distribution of these COVID-19 associated genes (Figure 3). COVID-19 associated disease genes are associated with nervous system disease (772), gastrointestinal system disease (513), endocrine system disease (498), musculoskeletal system disease (498), skin & connective tissue disease (479) and immune & inflammatory disease (431) (Figure 3C). Among these six organ systems affected by COVID genes, we drill down into each disease branch and list 10 granular disease terms selected by gene counts from the Comparison Heat Map in the Gene Annotator tool (Table 3). Among COVID-19 genes, over sixty percent (772/1257) are also involved in nervous system diseases. Out of 772 COVID/nervous system disease genes, over 650 are involved in the central nervous system followed by sensory system, neurologic manifestation, and neurodegenerative disease. Among gastrointestinal system diseases and endocrine system diseases, liver diseases, including liver neoplasms and cancers, show high prevalence. This correlates with the results of enrichment analysis where liver diseases were highly enriched by both tools. There are only 431 genes involved in the immune & inflammatory disease branch; however, diseases related to immune and inflammatory are present in all the six organ systems listed in Table 3. They are autoimmune disease of the nervous system, autoimmune disease of gastrointestinal tract, autoimmune disease of endocrine system, autoimmune disease of musculoskeletal system and dermatitis.

### 3.6. Gene Ontology Enrichment of COVID Genes

The Gene Ontology enrichment patterns of COVID genes were examined using the MOET tool. The Biological Process (BP) enrichment list is heavily concentrated in the ‘immune system process’ branch which includes immune response, immune effector process, leukocyte activation, and their child terms (Table 4 and Appendix A). Another highly represented branch is ‘response to stimulus’ where the fourth enriched term ‘defense response’ resides. The cellular component annotations of COVID genes are highly enriched in the branches of ‘immunoglobulin complex,’ ‘extracellular region’, ‘membrane’ and ‘cell periphery’ (Table 4 and Appendix A). Most of the top 40 Molecular Function (MF) terms are either ‘binding’ or its child terms such as antigen binding, protein binding or carbohydrate binding in the binding branch. The rest of the terms are ‘molecular function regulator,’ and its child terms under the branch (Table 4 and Appendix A).

## 4. Discussion

COVID-19 was declared a pandemic in March 2020, less than three months after its first identification. The whole world has allocated resources to study COVID-19 with the hope to find preventive and therapeutic measures to control the pandemic. The immense efforts in studying COVID-19 have produced over 280,000 publications and related datasets. From these available resources, the RGD team was able to curate and integrate COVID-19 associated data and to release the COVID-19 Disease Portal just four months later in July 2020 with regular updates. In this manuscript, COVID-19 genes were taken from the portal and analyzed with tools developed in-house and other publicly available tools.

Most of the curated disease genes were curated with evidence codes HEP or IEP, which identify changes in the gene expression pattern during COVID-19 disease (Table 1). These genes can serve as biomarkers to monitor the disease course and devise treatment plans. Currently, immune/inflammatory cytokine patterns are known to be useful in predicting disease progression [10], and treatments based on blocking excessive cytokine release have been proposed as a treatment regime [26]. The COVID-19 associated genes were examined by their distribution among anatomical entities and the over-represented diseases. Over sixty percent (772/1257) of the COVID-19 disease genes are also involved in ‘nervous system diseases’ and ‘autoimmune disease of central nervous system’ was among the top 40 enriched diseases. According to the breakdowns of nervous system disease gene counts in Table 3, the central nervous system, peripheral nervous system and sensory system are affected by the COVID genes and activation of autoimmunity in these systems is the major disease mechanism. More than likely, the central nervous system is the most significant target since its autoimmune disease is among one of the top 40 most enriched diseases. These results suggest that immune/inflammatory attacks on the nervous system play roles in the neurological manifestations such as loss of smell, headache, nausea, and impaired consciousness experienced by some COVID-19 patients [8,25,27]. The immune system, which is important to fight off infection, when dysregulated, becomes destructive and causes severe disease complications such as ‘cytokine storm’ in severe COVID cases [10,28]. Our enrichment analysis shows that several immune/inflammation diseases were on the top 40 list, and immune dysregulation was implicated in all the six organ systems examined in Table 3. The involvement of COVID-19 genes in the immune system was further confirmed by Gene Ontology enrichment profiles as shown in Table 4. All three aspects pointed to terms associated with the immune system. It has been shown previously that different disease gene sets exhibit unique GO enrichment profiles which reflect the unique pathophysiology of the disease [29]. Our analysis of COVID-19 disease genes suggests that dysregulation of immune function is a common mechanism affecting pathogenesis of COVID-19 disease, especially in the severe cases where multiple organ systems are affected.

COVID-19 was first identified as a respiratory disease and in severe cases, caused serious lung injury, particularly the high ACE 2 expression type II alveoli, and resulting respiratory distress [2,30]. In addition to lungs, ACE2 has a broad distribution in tissues and vasculature and its expression provides viral entry to the organ system. Viral damage to liver, brain, and kidney has been documented in biopsies from patients [2,5,6]. However, as a newly evolving disease, whether direct SARS-CoV2 entry through ACE2 leads to multiple organ damages is not clear. The unexpected finding of liver diseases as being highly enriched in genes associated with COVID-19 by two enrichment tools could point to an important direction to understand the pathogenesis of COVID-19. It has been shown that liver organoids express viral receptor ACE2 and can be infected by SARS-Co-V2 and become a replication reservoir for the virus [2]. The percentage of liver injury associated with COVID-19 patients varied among patient groups [31]; however, liver damages seemed to link to severe cases and poor disease outcome [32,33]. The liver damage caused by SARS-CoV2 infectionis attributed to several mechanisms such as augmented expression of the viral receptor ACE2 during diseases, uncontrolled immune cell infiltration, and cytokine storms after infection in the hepatobiliary system [32,33,34]. These mechanisms might also be involved in multiple organ injuries observed COVID-19 patients. What is unique in liver injury and COVID is its association with unbalanced coagulation control. It has been shown that COVID-19 patients exhibited a hypercoagulable state and this condition is related to impaired liver function resulted from liver injury [34]. Since the liver is the major organ producing coagulation factors [35], damage to liver would aggravate coagulation control thus exacerbating liver failure or even multiple organ failure in the severe cases. However, alteration in hemostasis control could be secondary to cytokine dysregulation since our enrichment analyses did not show overrepresentation of blood coagulation diseases. Here, we performed a detailed analysis of COVID-19 genes by their association with organ systems, disease enrichment, and GO enrichment using the resources available at RGD. Some of the results confirm with the clinical features of COVID-19 such as the involvement of nervous system and immune system in the disease. Additionally, the enrichment results point out the link between COVID-19 and liver diseases. As a globally accessible disease bioinformatic resource, RGD strives to provide researchers with utilities to resolve the complexity of disease research. During the challenging time of the COVID-19 pandemic, producing, organizing, and integrating data sets related to the disease is a timely contribution to the disease research community.

## Figures and Tables

**Figure 1 genes-13-02304-f001:**
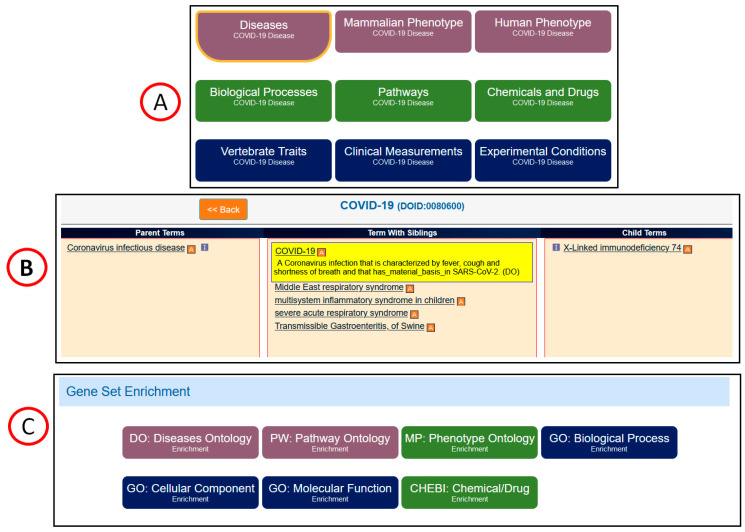
The COVID-19 Disease Portal landing page. (**A**). Annotations made to COVID-19 genes are organized into nine categories displayed in the COVID-19 Disease Portal. (**B**). The customized disease ontology browser showing terms associated with COVID-19 disease genes is shown at the middle. (**C**). Seven ontology enrichment analyses allow users to send COVID-19 disease genes to the MOET tool for gene set enrichment analysis.

**Figure 2 genes-13-02304-f002:**
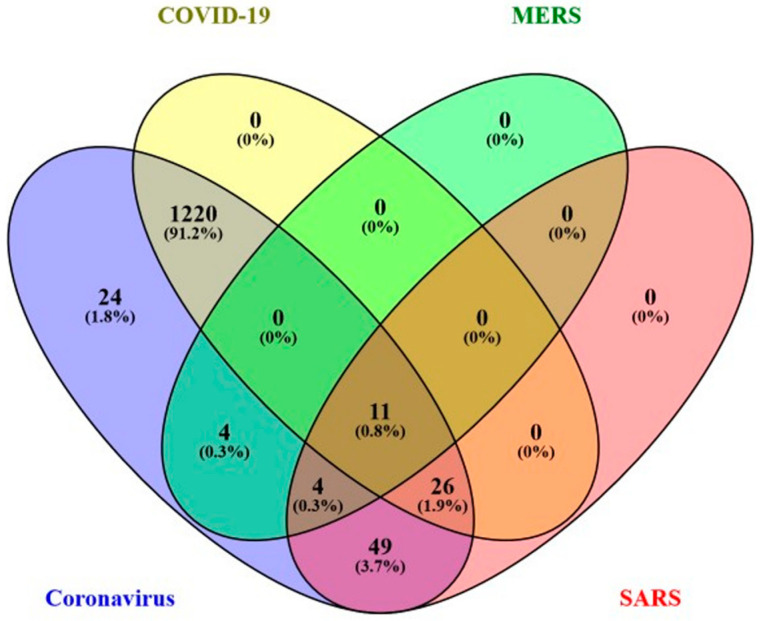
The coronavirus disease gene distribution among the parent term (Coronavirus infectious disease) and its three child terms: COVID-19, Middle East respiratory syndrome (MERS) and severe acute respiratory syndrome (SARS). The numbers in each area represent the gene count of that section and the percentage to all the coronavirus infectious disease genes. There are 1257 genes associated with COVID-19, 19 genes associated with MERS and 90 genes with SARS, totaling 1338 coronavirus infectious disease genes on display.

**Figure 3 genes-13-02304-f003:**
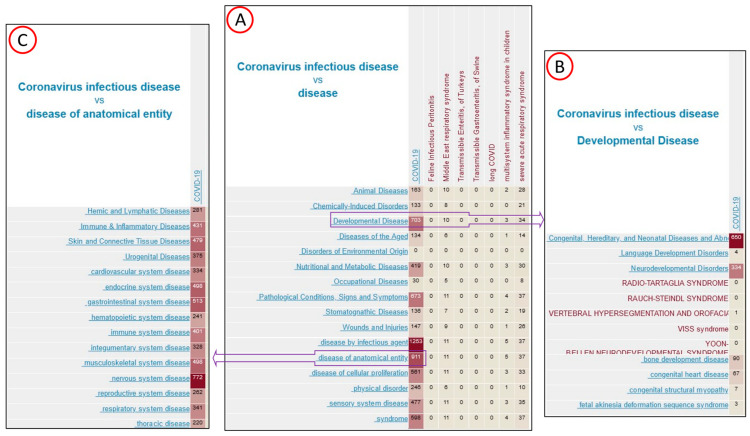
The disease term Comparison Heat Map visualized in the Gene Annotator tool. (**A**). COVID disease genes were visualized by their association with high level disease terms. (**B**). COVID genes associated with developmental disease were expanded to show their association with more granular terms under the branch. (**C**). COVID genes associated with disease of anatomical entity were expanded to show their association with more granular terms under the branch.

**Table 1 genes-13-02304-t001:** Evidence code analysis of COVID-19 disease annotations and associated genes.

A. Evidence Code (ECO) Distribution			B. Unique ECO among Genes	
ECO Type	Annotation Count	Gene Count	Gene/Unique ECO	Gene Count
IAGP	21	19	1	221
IDA	4	3	2	28
IMP	3	3	3	7
IEP	145	63	4	1
HEP	1195	1175		
EXP	37	37		
ISO	2	2		
Total	1407	1257		

A. The breakdown of 1407 COVID-19 disease annotations and 1257 associated disease genes according to evidence code types. B. The breakdown of gene counts by their association with unique evidence codes.

**Table 2 genes-13-02304-t002:** A. Top 40 enriched disease terms identified by MOET and Set Analyzer. * The Bonferroni corrected *p*-values are shown; #, the number of the COVID-19 related disease gene annotated with the disease term and its child term; &, the number of genes in the reference genome annotated with the disease term and its child terms. MOET, the Multi Ontology Enrichment Tool, https://rgd.mcw.edu/rgdweb/enrichment/start.html (accessed on 1 June 2022), developed by RGD. Set Analyzer, the enrichment tool developed at CTD (http://ctdbase.org/tools/analyzer.go (accessed on 1 June 2022)). B. * The Bonferroni corrected *p*-values are shown; #, the number of the COVID-19 related disease gene annotated with the disease term and its child term; &, the number of genes in the reference genome annotated with the disease term and its child terms. Set Analyzer, the enrichment tool developed at CTD (http://ctdbase.org/tools/analyzer.go (accessed on 1 June 2022)).

Top 40 Enriched Disease Ontology Terms from MOET and Set Analyzer				
	**A. MOET**			
**Rank**	**Disease Term (ID)**	***p* ***	**Count #**	**Ref Count &**
1	Respiratory Tract Infections (DOID:9008680)	1.92 × 10^−40^	135	553
2	liver disease (DOID:409)	2.16 × 10^−35^	335	2638
3	hepatocellular carcinoma (DOID:684)	9.75 × 10^−34^	161	853
4	liver carcinoma (DOID:686)	1.13 × 10^−33^	161	854
5	hepatobiliary disease (DOID:3118)	1.49 × 10^−32^	337	2745
6	liver cancer (DOID:3571)	5.51 × 10^−32^	164	908
7	hepatobiliary system cancer (DOID:0080355)	7.52 × 10^−32^	171	975
8	Adenocarcinoma (DOID:299)	1.31 × 10^−28^	228	1617
9	Liver Neoplasms (DOID:9007188)	1.05 × 10^−27^	176	1100
10	rheumatic disease (DOID:1575)	1.18 × 10^−27^	133	698
11	autoimmune disease of musculoskeletal system (DOID:0060032)	7.15 × 10^−27^	156	922
12	allergic disease (DOID:1205)	3.50 × 10^−26^	122	624
13	pneumonia (DOID:552)	1.18 × 10^−25^	79	292
14	nephritis (DOID:10952)	7.85 × 10^−25^	91	388
15	bacterial infectious disease (DOID:104)	4.20 × 10^−24^	130	728
16	respiratory allergy (DOID:0060496)	2.25 × 10^−23^	90	397
17	Glomerulonephritis (DOID:2921)	2.83 × 10^−23^	82	337
18	Wounds and Injuries (DOID:9001600)	3.76 × 10^−23^	147	906
19	Immediate Hypersensitivity (DOID:9002850)	8.83 × 10^−23^	99	477
20	Bacterial Infections and Mycoses (DOID:9004384)	1.24 × 10^−22^	149	936
21	Glomerular Diseases (DOID:9000104)	1.26 × 10^−22^	82	344
22	Inflammation (DOID:9005372)	3.57 × 10^−22^	297	2605
23	parasitic protozoa infectious disease (DOID:2789)	3.64 × 10^−22^	56	170
24	rheumatoid arthritis (DOID:7148)	3.65 × 10^−22^	94	444
25	obstructive lung disease (DOID:2320)	6.72 × 10^−22^	96	464
26	carcinoma (DOID:305)	1.02 × 10^−21^	346	3235
27	arthritis (DOID:848)	2.80 × 10^−21^	140	875
28	bone inflammation disease (DOID:3342)	5.18 × 10^−21^	143	910
29	lung disease (DOID:850)	9.56 × 10^−21^	248	2066
30	parasitic infectious disease (DOID:1398)	9.89 × 10^−21^	77	327
31	Orthomyxoviridae Infections (DOID:9001499)	8.07 × 10^−20^	49	144
32	asthma (DOID:2841)	8.18 × 10^−20^	80	361
33	hepatitis (DOID:2237)	1.05 × 10^−19^	71	293
34	Thoracic Injuries(DOID:9001954)	1.35 × 10^−19^	68	272
35	bronchial disease (DOID:1176)	1.57 × 10^−19^	145	962
36	lower respiratory tract disease (DOID:0050161)	1.92 × 10^−19^	248	2110
37	influenza (DOID:8469)	2.23 × 10^−19^	48	141
38	autoimmune disease of central nervous system (DOID:0060004)	4.09 × 10^−19^	72	307
39	respiratory system disease (DOID:1579)	4.16 × 10^−19^	343	3306
40	Lung Injury (DOID:9000310)	1.17 × 10^−18^	66	267
	**B. Set Analyzer**			
**Rank**	**MESH Term (ID)**	** *p* ** *****	**Count #**	**Ref Count &**
1	Neoplasms (MESH:D009369)	5.22 × 10^−126^	387	3777
2	Digestive System Diseases (MESH:D004066)	1.52 × 10^−117^	325	2810
3	Liver Diseases (MESH:D008107)	1.22 × 10^−114^	272	1974
4	Neoplasms by Site (MESH:D009371)	4.75 × 10^−99^	310	2978
5	Immune System Diseases (MESH:D007154)	2.63 × 10^−91^	195	1219
6	Carcinoma (MESH:D002277)	1.28 × 10^−84^	196	1341
7	Digestive System Neoplasms (MESH:D004067)	8.85 × 10^−79^	197	1461
8	Respiratory Tract Diseases (MESH:D012140)	2.39 × 10^−77^	171	1099
9	Carcinoma, Hepatocellular (MESH:D006528)	7.71 × 10^−76^	122	515
10	Adenocarcinoma (MESH:D000230)	1.00 × 10^−74^	163	1028
11	Liver Neoplasms(MESH:D008113)	1.47 × 10^−72^	136	707
12	Skin and Connective Tissue Diseases (MESH:D017437)	7.70 × 10^−72^	200	1650
13	Lung Diseases (MESH:D008171)	1.13 × 10^−71^	151	910
14	Urogenital Diseases (MESH:D000091642)	1.08 × 10^−70^	226	2135
15	Nervous System Diseases(MESH:D009422)	3.17 × 10^−69^	253	2695
16	Female Urogenital Diseases and Pregnancy Complications (MESH:D005261)	1.42 × 10^−65^	185	1535
17	Respiratory Tract Infections (MESH:D012141)	2.29 × 10^−65^	74	177
18	Infections (MESH:D007239)	1.62 × 10^−64^	114	543
19	Skin Diseases (MESH:D012871)	2.51 × 10^−63^	170	1336
20	Pneumonia (MESH:D011014)	1.51 × 10^−62^	57	93
21	Female Urogenital Diseases (MESH:D052776)	3.36 × 10^−62^	172	1392
22	Pneumonia, Viral(MESH:D011024)	1.02 × 10^−57^	38	38
23	Fibrosis (MESH:D005355)	4.37 × 10^−57^	142	1020
24	Male Urogenital Diseases (MESH:D052801)	3.56 × 10^−56^	172	1528
25	COVID-19 (MESH:D000086382)	3.99 × 10^−56^	37	37
26	Urologic Diseases (MESH:D014570)	6.12 × 10^−55^	128	852
27	Autoimmune Diseases (MESH:D001327)	1.01 × 10^−53^	104	552
28	Virus Diseases (MESH:D014777)	5.58 × 10^−48^	78	324
29	Kidney Diseases (MESH:D007674)	2.34 × 10^−47^	108	694
30	Musculoskeletal Diseases (MESH:D009140)	5.70 × 10^−47^	159	1529
31	Hypersensitivity (MESH:D006967)	4.50 × 10^−45^	76	331
32	Nephritis (MESH:D009393)	6.61 × 10^−45^	52	126
33	Gastrointestinal Diseases (MESH:D005767)	2.78 × 10^−44^	129	1072
34	Liver Cirrhosis(MESH:D008103)	3.25 × 10^−44^	118	897
35	Vascular Diseases (MESH:D014652)	5.72 × 10^−44^	122	965
36	Cardiovascular Diseases (MESH:D002318)	4.13 × 10^−42^	151	1515
37	Connective Tissue Diseases (MESH:D003240)	5.25 × 10^−41^	88	521
38	Liver Cirrhosis, Experimental (MESH:D008106)	6.80 × 10^−41^	106	777
39	Central Nervous System Diseases (MESH:D002493)	3.99 × 10^−40^	138	1330
40	Glomerulonephritis (MESH:D005921)	4.90 × 10^−40^	46	109

**Table 3 genes-13-02304-t003:** COVID disease gene distribution within organ systems.

Nervous System Disease (772)	Gastrointestinal System Disease (513)	Endocrine System Disease (498)	Musculoskeletal System Disease (498)	Skin and Connective Tissue Disease (479)	Immune & Inflammatory Disease (431)
central nervous system disease (676)	gastrointestinal disease (401)	liver disease (335)	connective tissue disease (376)	bone disease (270)	primary innunodeficiency disease (348)
sensory system disease (477)	digestive system neoplasm (339)	endocrine gland neoplasm (301)	bone disease (270)	connective tissue neoplasm (161)	autoimmune disease (246)
neurologic Manifestation (356)	liver disease (335)	diabetes Mellitus (134)	neuromusclular disease (174)	breast disase (161)	lymphatic system disease (176)
neurodegenerative disease (278)	intestinal disease (248)	pancreas disease (119)	musculoskeletal annormalities (157)	rheumatic disease (123)	rheumatic disease (133)
peripheral nervous system disease (232)	mouth disease (120)	gonaldal disease (109)	autoimmune disease of musculoskeletal system (156)	genetic skin disease (119)	allergic disease (121)
nervous system neoplasm (148)	stomach disease (99)	parathyoid gland disease (88)	joint disease (156)	interstitial lung disease (85)	immunoproliferative disorders (127)
nervous system malformation (122)	gastroenteritis (97)	autoimmune disease of endcrine system (61)	muscular disease (113)	dermatitis (78)	immune ststem cancer (99)
nervous system trauma (98)	biliary tract disease (66)	thyroid gland disease (53)	musculoskeletal system cancer (93)	collagen disease (65)	gastroenteritis (97)
autoimmune disease of the nervous system (83)	esophageal disease (56)	dwarfism (29)	jaw disease (37)	eczematous skin disease (57)	dermatitis (78)
congenital nervous system abnormality (75)	autoimmune disease of gastrointestinal tract (55)	adrenal gland disease (24)	musculoskeketal system benign neoplasm (10)	infectious skin disease (62)	pneumonia (79)

The six organ system diseases associated with highest number of COVID-19 disease genes were drilled down to more granular diseases within the branch. The number in parentheses next to the disease term shows the number of COVID-19 associated genes that are associated the disease term. These numbers were taken from the Comparison Heat Map in the Gene Annotator tool. The 6 organ system diseases with highest disease gene counts were selected from Figure 3C.

**Table 4 genes-13-02304-t004:** Top 40 enriched gene ontology terms identified by MOET.

	**A. Biological Process**			
**Rank**	**Term (ID)**	***p* ***	**Count #**	**Ref Count &**
1	immune system process (GO:0002376)	5.93 × 10^−76^	422	3246
2	immune response (GO:0006955)	3.97 × 10^−74^	331	2154
3	adaptive immune response (GO:0002250)	1.28 × 10^−65^	186	805
4	defense response (GO:0006952)	6.89 × 10^−52^	286	2067
5	response to external stimulus (GO:0009605)	6.18 × 10^−45^	382	3525
6	biological process involved in interspecies interaction between organisms (GO:0044419)	8.29 × 10^−42^	269	2093
7	response to stimulus (GO:0050896)	3.90 × 10^−39^	752	10120
8	response to other organism (GO:0051707)	1.58 × 10^−38^	249	1925
9	response to biotic stimulus (GO:0009607)	1.93 × 10^−38^	253	1977
10	response to external biotic stimulus (GO:0043207)	2.09 × 10^−38^	249	1928
11	response to stress (GO:0006950)	2.16 × 10^−38^	443	4670
12	defense response to other organism (GO:0098542)	3.13 × 10^−38^	198	1327
13	complement activation classical pathway (GO:0006958)	3.51 × 10^−38^	59	123
14	leukocyte activation (GO:0045321)	1.03 × 10^−37^	183	1171
15	humoral immune response (GO:0006959)	1.11 × 10^−37^	101	394
16	humoral immune response mediated by circulating immunoglobulin (GO:0002455)	6.29 × 10^−37^	61	138
17	adaptive immune response based on somatic recombination of immune receptors built from immunoglobulin superfamily domains (GO:0002460)	8.14 × 10^−37^	103	418
18	complement activation (GO:0006956)	5.99 × 10^−36^	63	153
19	cell activation (GO:0001775)	1.53 × 10^−35^	194	1337
20	innate immune response (GO:0045087)	3.71 × 10^−35^	166	1035
21	leukocyte mediated immunity (GO:0002443)	4.46 × 10^−35^	113	520
22	lymphocyte mediated immunity (GO:0002449)	6.42 × 10^−35^	99	405
23	positive regulation of immune system process (GO:0002684)	1.03 × 10^−34^	172	1108
24	immune effector process (GO:0002252)	3.67 × 10^−34^	140	784
25	immunoglobulin mediated immune response (GO:0016064)	5.01 × 10^−34^	74	230
26	B cell mediated immunity (GO:0019724)	2.63 × 10^−33^	74	235
27	regulation of immune system process (GO:0002682)	1.57 × 10^−32^	221	1729
28	phagocytosis recognition (GO:0006910)	4.59 × 10^−31^	54	128
29	positive regulation of B cell activation (GO:0050871)	5.68 × 10^−31^	63	180
30	regulation of B cell activation (GO:0050864)	1.24 × 10^−30^	72	241
31	regulation of leukocyte activation (GO:0002694)	5.52 × 10^−30^	132	771
32	positive regulation of immune response (GO:0050778)	1.3 × 10^−29^	123	687
33	phagocytosis engulfment (GO:0006911)	6.23 × 10^−29^	56	150
34	cell surface receptor signaling pathway (GO:0007166)	1.92 × 10^−28^	319	3186
35	plasma membrane invagination (GO:0099024)	2.15 × 10^−28^	57	159
36	response to bacterium (GO:0009617)	2.31 × 10^−28^	161	1114
37	inflammatory response (GO:0006954)	3.21 × 10^−28^	149	983
38	lymphocyte activation (GO:0046649)	3.59 × 10^−28^	148	973
39	membrane invagination (GO:0010324)	6.48 × 10^−28^	58	168
40	positive regulation of response to stimulus (GO:0048584)	1.09 × 10^−27^	272	2545
	**B. Cellular Component**			
**Rank**	**Term (ID)**	***p* ***	**Count #**	**Ref Count &**
1	immunoglobulin complex (GO:0019814)	2.22 × 10^−95^	114	186
2	extracellular space (GO:0005615)	2.41 × 10^−51^	409	3812
3	external side of plasma membrane (GO:0009897)	6.87 × 10^−42^	124	558
4	extracellular region (GO:0005576)	8.80 × 10^−42^	453	4820
5	immunoglobulin complex circulating (GO:0042571)	7.25 × 10^−39^	51	90
6	cell surface (GO:0009986)	7.98 × 10^−35^	172	1147
7	side of membrane (GO:0098552)	9.48 × 10^−34^	140	822
8	cell periphery (GO:0071944)	7.43 × 10^−26^	518	6662
9	blood microparticle (GO:0072562)	6.74 × 10^−22^	48	153
10	plasma membrane (GO:0005886)	6.93 × 10^−22^	476	6160
11	vesicle (GO:0031982)	8.59 × 10^−20^	362	4363
12	extracellular exosome (GO:0070062)	4.55 × 10^−16^	214	2254
13	membrane (GO:0016020)	1.96 × 10^−14^	674	10362
14	extracellular vesicle (GO:1903561)	3.25 × 10^−14^	215	2354
15	extracellular organelle (GO:0043230)	4.13 × 10^−14^	215	2359
16	extracellular membran × 10−bounded organelle (GO:0065010)	4.13 × 10^−14^	215	2359
17	intracellular vesicle (GO:0097708)	5.91 × 10^−12^	229	2684
18	cytoplasmic vesicle (GO:0031410)	9.94 × 10^−12^	228	2681
19	secretory granule lumen (GO:0034774)	3.68 × 10^−10^	52	324
20	cytoplasmic vesicle lumen (GO:0060205)	5.33 × 10^−10^	52	327
21	vesicle lumen (GO:0031983)	6.80 × 10^−10^	52	329
22	lytic vacuole (GO:0000323)	9.38 × 10^−9^	89	801
23	lysosome (GO:0005764)	9.38 × 10^−9^	89	801
24	protein-containing complex (GO:0032991)	9.66 × 10^−9^	450	6667
25	vacuole (GO:0005773)	3.28 × 10^−8^	95	900
26	secretory vesicle (GO:0099503)	6.92 × 10^−8^	115	1190
27	secretory granule (GO:0030141)	8.02 × 10^−8^	97	942
28	IgG immunoglobulin complex (GO:0071735)	7.01 × 10^−6^	8	11
29	endomembrane system (GO:0012505)	9.94 × 10^−6^	340	5003
30	vacuolar lumen (GO:0005775)	1.18 × 10^−4^	28	176
31	endocytic vesicle (GO:0030139)	1.23 × 10^−4^	45	372
32	collagen-containing extracellular matrix (GO:0062023)	1.60 × 10^−4^	52	464
33	membrane raft (GO:0045121)	1.90 × 10^−4^	49	428
34	cellular anatomical entity (GO:0110165)	1.95 × 10^−4^	1072	20217
35	membrane microdomain (GO:0098857)	2.04 × 10^−4^	49	429
36	tertiary granule lumen (GO:1904724)	5.41 × 10^−4^	14	55
37	extracellular matrix (GO:0031012)	1.06 × 10^−3^	60	602
38	external encapsulating structure (GO:0030312)	1.18 × 10^−3^	60	604
39	cytoplasm (GO:0005737)	1.26 × 10^−3^	729	12592
40	specific granule lumen (GO:0035580)	2.57 × 10^−3^	14	62
	**C. Molecular Function**			
**Rank**	**Term (ID)**	***p* ***	**Count #**	**Ref Count &**
1	antigen binding (GO:0003823)	9.64 × 10^−65^	96	201
2	immunoglobulin receptor binding (GO:0034987)	1.43 × 10^−36^	51	95
3	signaling receptor binding (GO:0005102)	9.52 × 10^−21^	194	1748
4	cytokine activity (GO:0005125)	1.75 × 10^−14^	51	240
5	cytokine receptor binding (GO:0005126)	3.61 × 10^−13^	55	294
6	immune receptor activity (GO:0140375)	3.27 × 10^−11^	37	160
7	chemokine activity (GO:0008009)	4.59 × 10^−10^	20	50
8	signaling receptor activator activity (GO:0030546)	1.85 × 10^−9^	70	525
9	receptor ligand activity (GO:0048018)	2.34 × 10^−9^	69	516
10	chemokine receptor binding (GO:0042379)	5.32 × 10^−9^	23	75
11	signaling receptor regulator activity (GO:0030545)	7.91 × 10^−9^	73	577
12	protein binding (GO:0005515)	2.21 × 10^−8^	895	15279
13	cytokine binding (GO:0019955)	2.29 × 10^−8^	32	151
14	identical protein binding (GO:0042802)	3.85 × 10^−8^	200	2394
15	cytokine receptor activity (GO:0004896)	7.13 × 10^−8^	25	99
16	chemokine binding (GO:0019956)	8.63 × 10^−6^	13	33
17	binding (GO:0005488)	8.63 × 10^−5^	985	17676
18	growth factor activity (GO:0008083)	1.00 × 10^−4^	28	167
19	C-C chemokine binding (GO:0019957)	2.76 × 10^−4^	10	24
20	heparin binding (GO:0008201)	2.97 × 10^−4^	29	186
21	growth factor binding (GO:0019838)	3.31 × 10^−4^	26	156
22	enzyme binding (GO:0019899)	1.75 × 10^−3^	177	2362
23	C-C chemokine receptor activity (GO:0016493)	2.19 × 10^−3^	9	23
24	CXCR chemokine receptor binding (GO:0045236)	2.70 × 10^−3^	8	18
25	CCR chemokine receptor binding (GO:0048020)	2.94 × 10^−3^	13	51
26	G protein-coupled chemoattractant receptor activity (GO:0001637)	7.23 × 10^−3^	9	26
27	chemokine receptor activity (GO:0004950)	7.23 × 10^−3^	9	26
28	glycosaminoglycan binding (GO:0005539)	7.48 × 10^−3^	33	264
29	protein homodimerization activity (GO:0042803)	1.00 × 10^−2^	69	749
30	growth factor receptor binding (GO:0070851)	1.13 × 10^−2^	23	154
31	G protein-coupled receptor binding (GO:0001664)	1.31 × 10^−2^	38	333
32	molecular function regulator (GO:0098772)	3.08 × 10^−2^	153	2082
33	exogenous protein binding (GO:0140272)	3.91 × 10^−2^	15	82
34	kinase binding (GO:0019900)	4.78 × 10^−2^	77	904
35	cell adhesion molecule binding (GO:0050839)	4.97 × 10^−2^	55	585
36	protease binding (GO:0002020)	6.58 × 10^−2^	22	160
37	sulfur compound binding (GO:1901681)	7.12 × 10^−2^	34	307
38	carbohydrate derivative binding (GO:0097367)	1.10 × 10^−1^	168	2383
39	virus receptor activity (GO:0001618)	1.38 × 10^−1^	14	81
40	protein-containing complex binding (GO:0044877)	1.49 × 10^−1^	122	1642

* The Bonferroni corrected *p*-values are shown; #, the number of the COVID-19 related disease gene annotated with the GO term and its child term; &, the number of genes in the reference genome annotated with the GO term and its child terms.

## Data Availability

The datasets and computer tools used to generate and/or analyze the results during the current study are either publicly available or available from the corresponding authors on request.

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
