# Peer review of "Ontological Analysis of Coronavirus Associated Human Genes at the COVID-19 Disease Portal"

_genes, 2022, doi:10.3390/genes13122304_

Round 1

Reviewer 1 Report

Overview

The authors present the COVID-19 Disease Portal as a way to leverage existing PubMed literature concerning COVID-19 enriched by ontologies, in the interest analyzing genes relevant to COVID-19, and related diseases.

Necessary Changes

While written well overall, the authors' results were difficult to parse at points, owing to too little exposition in some places. More specifically:

The authors gloss over several tools throughout the paper, making it challenging to understand motivations for and results of the current project. The authors should provide at least brief explanations (two sentences at most should suffice) for the following tools, why they were used, and what results were obtained:

Gene Annotator

MOET

CTD Set Analyzer

This is particularly pressing when - as the authors note - these tools provide rankings for enriched terms with different results. The authors observe, for example, that rankings between MOET and the Set Analyzer were different, but they do not explain why or what importance this difference has. I suspect more exposition of these tools would make clear why these different rankings emerged, and how that result bears on the authors' work.

The authors write that their results (311) might shed light on the mechanisms involved in neurological manifestations of COVID-19. I agree, but it is incumbent on the authors to say more about how, perhaps by venturing hypotheses of the sort discussed in the following paragraph on liver disease.

The discussion section would be benefitted if the authors compared their toolkit to other nearby tools, such as the web-based functional annotation tools associated with the Database for Annotation, Visualization, and Integrated Discovery (DAVID). The DAVID toolkit has been used, for example, in ontology term enrichment for genes associated with COVID-19 and acute kidney disease (https://ceur-ws.org/Vol-3073/paper15.pdf). Comparison between these strategies and toolkits would be informative, and perhaps strengthen the authors' results.

Good to Have Changes

Related, it would be informative if the authors would provide - alongside their hypotheses concerning the liver and COVID-19 - how the results of their analyses can rule out hypotheses for the observed phenomena, critical for newly evolving pathogens. Put another way, the authors might investigate whether there are hypotheses in the literature concerning liver damage by SARS-CoV-2, that seem improbable given their enrichment results.

The authors state that a prioritized disease gene list was constructed "as previously described" (103) where there is reference to a published article. The authors should describe the constructed gene list briefly here, rather than reference a description elsewhere.

Suggestions

The authors spend a lot of space outlining SARS-CoV-2 pathogenesis in the introduction. The exposition would be welcome if the details provided in this section were leveraged later in the paper. I suggest either trimming this discussion of pathogenesis, or clarifying how the details provided in this section inform, enhance, etc. the authors' analyses later in the paper.

The authors might consider adding the Coronavirus Infectious Disease Ontology, which alongside the Disease Ontology and the Gene Ontology, is part of the OBO Foundry of biomedical ontologies, though focused specifically on coronaviruses.

The authors write that they were able to curate and create the COVID-19 Disease Portal "in a timely manner" (299). I encourage the authors be more specific.

Grammar Changes

Line 176: Need space after "1257"

Lines 190-1: Should be "breakdowns of the anatomical entities…"

Line 195: Extra space after "genes"

Lines 201-4: Suggest rewriting the sentence, strikes me as ungrammatical.

Line 251: "650s" should be "650"

Line 281: Extra space before "COVID-19"; need space between "," and "the"

Line 289: Extra space before "COVID-19"; need space between "," and "the"

Line 309-10: Suggest rewriting the sentence, strikes me as ungrammatical.

Line 311: Remove "analysis"

Reviewer 2 Report

The authors present a COVID-19 Disease Portal to utilize information from the scientific literature. Illness Ontology enrichment analysis was done on the data in the portal and revealed that the COVID-19 gene set is strongly enriched with coronavirus infectious disease and associated conditions. Less connected disorders, such as hepatic and rheumatic diseases, were also considerably enhanced. I believe that the table containing the top 40 enhanced disease ontology keywords from MOET and Set Analyzer is a very useful resource for researchers. The work is properly organized and presented, the procedures are given clearly, and the findings are thoroughly addressed. The conclusion of the paper (The information in RGD’s COVID-19 disease portal can generate new hypotheses to potentiate novel therapies and prevention of acute and long-term complications of 28 COVID-19) is well supported by the data presented. This paper is an important addition to the literature. 

Author Response

No response is required.